# REPOST: Scalable Repository-Level Coding Environment Construction with Sandbox Testing

**Yiqing Xie**[1]  **Alex Xie**[1]  **Divyanshu Sheth**[1]  **Pengfei Liu**[2]  **Daniel Fried**[1]  **Carolyn Rosé**[1]
[1]Carnegie Mellon University  [2]Shanghai Jiao Tong University
yiqingxi@andrew.cmu.edu

## Abstract

We present REPOST, a scalable method to construct repository-level code generation environments that provide execution feedback, which can be used for both training and evaluation. Unlike existing works that require building the entire repository for execution, which is challenging for both human and LLMs, we provide execution feedback with *sandbox testing*, which isolates the target function and its dependencies to a separate script for testing. Sandbox testing reduces the complexity of external dependencies and enables constructing environments at a large scale. We use our method to construct REPOST-TRAIN, a large-scale train set with 7,415 functions from 824 repositories. Training with the execution feedback provided by REPOST-TRAIN leads to a performance gain of 5.5% Pass@1 on HumanEval and 3.5% Pass@1 on RepoEval. We also build an evaluation dataset, REPOST-EVAL, to showcase the potential of REPOST for live benchmark construction.[1]

## 1 Introduction

Code generation is a special NLP task that can benefit from execution-based feedback (Simon, 1963; Feng et al., 2020; Chen et al., 2021). With online coding platforms as a natural resource to build large-scale coding datasets with test cases, existing works demonstrate the effectiveness of execution-based signals in training data construction (Ni et al., 2024; Zhang et al., 2024; Liu et al., 2023a) and live evaluation (Jain et al., 2024a), which is updated over time to prevent contamination. However, models solely trained on algorithm problems cannot well generalize to repository-level (repo-level) code generation (Zhang et al., 2023; Jimenez et al., 2024), which aims to generate code for real, naturally occurring repositories and is well aligned with real-world software engineering practice.

It is non-trivial to build **execution-based** datasets for **repo-level** code generation at a **large scale**. One major challenge is to set up executable environments. Existing repo-level datasets typically conduct *integration testing* that executes test files inside the repositories, which requires building the entire repository (Jimenez et al., 2024; Jain et al., 2024b). As shown in prior research, this setup process is extremely challenging (Bogin et al., 2024), which may include installing external packages, determining execution context (e.g., relative import),

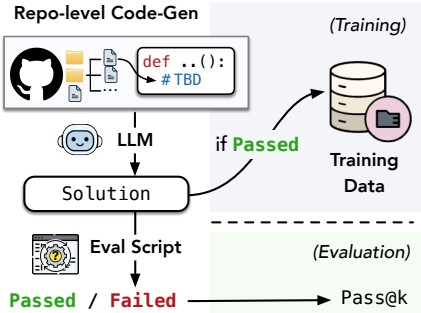

Figure 1: We can use the coding environments built by REPOST for training and evaluation. We first apply the code generation model to generate candidate solutions with the original repository as context. Then we evaluate the solutions by executing the evaluation script built by REPOST. For evaluation, we directly compute Pass@k scores. For training, we add all successful solutions to the train set and further finetune the model.

---

[1]Code and datasets available at https://repost-code-gen.github.io/.

debugging installation errors or runtime errors, etc. Hence, existing methods for coding environment construction either require huge manual effort (Zhang et al., 2023; Jimenez et al., 2024; Pan et al., 2024) or, when automated, suffer from low success rate (Jain et al., 2024b), limiting the scale of resulting datasets (see Table 1 for details).

In this work, instead of integration testing, we present REPOST, an **automated** framework to construct **Repo**-level coding environments with **S**andbox **T**esting. Specifically, given a function in a GitHub repository, we sandbox the function and its local dependencies to a separate script and generate tests with an LLM. To control the quality of the evaluation script, we iteratively resolve environment or runtime errors and improve test coverage. We also conduct execution-based, AST-based, and LLM-based quality checks and only keep examples where the functionality of the sandboxed function does not alter and the tests are valid, reasonable, and have high coverage. The resulting dataset allows us to obtain execution feedback in training and evaluation. As shown in Figure 1, the models generate the target function with the entire GitHub repository as context. We then use the evaluation script to obtain execution feedback.

Compared to integration testing used by previous datasets, we highlight the benefits of sandbox testing in constructing **scalable** coding environments: (1) In general, the external dependencies of a function are typically much simpler than a repository. By isolating the function and its local dependencies, we can execute the function by *only installing the necessary packages*. (2) If any execution error occurs, we can debug the separate scripts *without modifying the original repository*. This ensures that the datasets remain naturalistic: code generation models can still access the original, real-world repository.

With the high scalability of our framework, we construct coding environment datasets for both training and evaluation: REPOST-TRAIN and REPOST-EVAL. As shown in Table 1, to our knowledge, REPOST-TRAIN is currently the largest repo-level code generation dataset with execution support, with 7,415 functions sampled from 824 repositories. The large scale enables training on REPOST-TRAIN and evaluating on other benchmarks such as RepoEval (Zhang et al., 2023) or HumanEval (Chen et al., 2021). Due to its fully automated nature, REPOST also enables constructing live benchmarks to avoid contamination issues. We build REPOST-EVAL as an example, which contains 296 functions from 99 repositories.

Experiments show that by training on REPOST-TRAIN, we achieve performance gain on both algorithm problems in HumanEval (Chen et al., 2021) and repo-level tasks in RepoEval (Zhang et al., 2023) and REPOST-EVAL. For instance, we improve Qwen2.5-Coder by 5.5% Pass@1 on HumanEval and 3.5% Pass@1 on RepoEval. We also benchmark 12 code generation models on REPOST-EVAL. The best model (GPT-4o) only achieves 39.5 Pass@1, suggesting REPOST can construct challenging code generation benchmarks.

Note that REPOST-TRAIN provides coding tasks based on real-world repositories that have execution feedback, we would like to highlight that **REPOST can also be applied to coding agent** (Yang et al., 2024; Wang et al., 2025) **training and evaluation** in future works.

## 2 The REPOST Framework

The framework of REPOST is illustrated in Figure 2. After sampling GitHub repositories and extracting functions (§2.1), we sandbox each function and its local dependencies to a separate script (§2.2), generate tests (§2.3), and conduct quality control on the executability of the scripts, the functionality of the sandboxed function, and the test quality (§2.4). We provide all the prompts in §A.1. When we use our environments to train (§4) and evaluate (§5) code generation models, the models can still access the original GitHub repository, and we provide execution feedback by executing the evaluation scripts. To create user-friendly datasets, **we keep a shared Docker environment for all the evaluation scripts**.

### 2.1 Repository and Function Curation

We first randomly sample non-forked, MIT-licensed, Python GitHub Repositories with file sizes smaller than 10M. With our sandbox testing method, we are able to set up environments

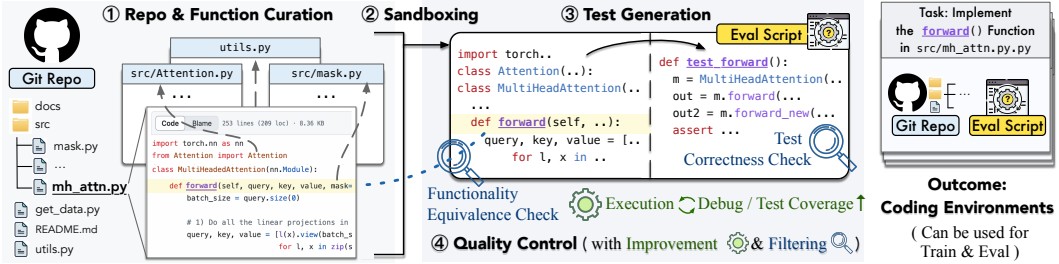

Figure 2: The REPOST coding environment construction framework. We sandbox the target function and its dependencies to a separate evaluation script for execution, which avoids building the entire repository. We design careful quality control strategies with iterative quality improvement and post-filtering. The outcome of REPOST is a set of executable repo-level coding environments, which can be used for training and evaluation.

for individual functions and do not need to build the entire repository. Hence, unlike previous works (Jain et al., 2024b), we do not need to filter out repositories without setup files (e.g., setup.py). The detailed repository statistics are provided in §3.

Then we extract functions from the repositories. To build the datasets in a docker with no access to GPUs or external services, we follow R2E (Jain et al., 2024b) and filter out functions associated with GPUs, cloud tasks, etc. by keywords. To balance the distribution of examples, we keep at most 30 functions for each repository. After this step, for the train set, we started from 1,000 repositories and obtained 17,448 functions from 851 repositories. For the evaluation set, we started from 200 repositories and obtained 1,043 functions from 139 repositories. In theory, we can further scale up the dataset with more starting repositories.

## 2.2 Sandboxing: Key to Environment Setup

It is non-trivial to create coding environments that provide correct execution feedback for the implementation of GitHub functions. Existing methods typically provide execution feedback with integration testing, which creates test files that import the target function from the codebase. Executing such tests requires building the entire repository, which generally requires a complicated environment and is challenging for both human and LLMs.

With the observation that the external dependencies of a function are typically much simpler than a repository, we tackle this challenge with **sandbox testing**, where we create a separate script containing the target function with the **exact same functionality** as the original one and the context that supports its executability. By isolating the function and its local dependencies, we can execute the function by only installing the necessary packages.

**Sandboxing based on Local Dependencies**. The major challenge of sandboxing is to ensure the executability of the function. While a standalone function can be directly copied to a separate

```
Orig Function

def API_call(prompt: str) -> str:
    return API_client.generate(
        model=API_MODEL, messages=[...]
    )
```

```
Eval Script

class Mock_API(object):
    ...
    def generate(model, messages):
        return "mock_" + messages[0]['content']

API_MODEL = "model_name"
API_client = Mock_API("api_key")
def API_call(prompt: str) -> str:
    return API_client.generate(
        model=API_MODEL, messages=[...]
    )
...
```

Figure 3: An example where the LLM successfully creates a mock class, Mock_API, to replace real external API calls. In this way, while the function body of the target function API_call remains exactly the same as in the original codebase, it can be executed without making real API calls.

script and executed, a typical GitHub function may depend on other modules, classes, or global variables in the same repository. Thus, we leverage the call graph to extract such

local dependencies that the target function directly or indirectly calls. Then we combine all the code fragments into the context and prompt an LLM (e.g., GPT-4o) to aggregate all the code fragments into a single script, with as little editing as possible.

**Sandboxing External APIs and Files**. Even if all the dependencies are presented, it is still nontrivial to execute the sandboxed script if it requires external API, databases, files, etc. We explicitly prompt the LLM to create mock connections for any external API and create strings or write example files to a specific directory for file reading. Figure 3 presents a successful case of sandboxing with mock APIs. We provide examples of the generated sandboxed scripts in Figures 5 to 10.

**Functionality Equivalence Control**. In addition to executability, another challenge of sandboxing is to ensure that the target function's functionality does not alter. We first conduct a list of sanity checks on the function name, length of scripts, etc. (see Table 11 for details), and re-generate examples that do not satisfy the requirements. We also have a final quality control step (in §2.4) that compares the functionality of the sandboxed and original functions. The precision of quality control is further verified in our human study in §3.1.

## 2.3 Test Generation

**Equivalence Testing**. We generate synthetic tests in the sandboxed script to evaluate the correctness of the generated code. Specifically, we prompt the LLM to (1) generate a set of test inputs to the target function and (2) conduct equivalence testing that checks whether the function generated in evaluation has the same behavior as the sandboxed function (i.e., the "ground-truth" implementation). Compared to traditional methods that specify the I/O examples in the test cases (Chen et al., 2021), equivalence testing does not need to predict the expected outputs and is more feasible for LLMs (Jain et al., 2024b).

**Test Generation with Mock APIs and Files**. We observe that the LLM is able to generate tests with the mock classes created in the sandboxing step as context. As shown in Figures 8 to 10, we create mock class instances as the test inputs and still ensure that the function body of the sandboxed function remains the same as the original function.

**Test Quality Control**. Similarly to the sandboxing step, for quality control purposes, we conduct a series of sanity checks, such as requiring the test function to call the target function and have at least 3 assertions (see Table 12 for the full list). We further check the coverage and correctness of the tests in the final quality control step (§2.4),

## 2.4 Quality Control and Filtering

We conduct quality control for the executability of the evaluation script, the functionality of the sandboxed function, and the test quality.

**Iterative Execution and Debugging**. In principle, if the model-generated function is exactly the same as the ground truth, the evaluation scripts should be successfully executed, with all the tests passed. Hence, we execute the examples sequentially in a docker. If there are any execution errors or if the ground truth implementation cannot pass any test cases, we provide the error message as the context and prompt an LLM to debug the evaluation script. Examples that still have errors after *k* execution-debugging iterations are filtered out.

We also dynamically install external packages during execution. During execution, if there are any `ModuleNotFound` errors, we extract the package names from the error message, run a `pip install` command, and execute the script again. In case the package name differs from the import name or the code requires a specific version of packages, we also allow the LLM to output package installation commands during debugging. In this way, we are able to install external packages for functions extracted from repositories without setup files.

**Iterative Test Coverage Improvement**. To ensure that the test functions cover the major functionality of the target functions, we further compute the branch coverage rate of all the evaluation scripts. If the branch coverage rate is lower than some threshold (we set 80% for

| Dataset | #Ex | #Repo | Repo? | AutoTest? |
|---|---|---|---|---|
| HumanEval | 164 | – | ✗ | ✗ |
| DS1000 | 1,000 | – | ✗ | ✗ |
| ClassEval | 100 | – | ✗ | ✗ |
| RepoEval-Func | 455 | 6 | ✓ | ✗ |
| SWE-Bench | 2,294 | 12 | ✓ | ✗ |
| CoderEval | 230 | 43 | ✓ | ✗ |
| DevEval | 1,874 | 117 | ✓ | ✗ |
| EvoCodeBench | 275 | 25 | ✓ | ✗ |
| SWE-Gym | 2,438 | 11 | ✓ | ✗ |
| R2E-Eval1 | 246 | 137 | ✓ | ✓ |
| R2E (Ours) | 744 | 123 | ✓ | ✓ |
| REPOST-TRAIN | 7,415 | 824 | ✓ | ✓ |
| REPOST-EVAL | 296 | 99 | ✓ | ✓ |

Table 1: Statistics of REPOST-TRAIN and REPOST-EVAL compared to existing **execution-based** code generation datasets. R2E (Our Input) applies the R2E method to the same set of input repositories as REPOST-TRAIN, but results in a smaller number of repositories and examples. "Repo?" and "Auto Test?" refer to the repo-level setting and automatically generated tests.

| Check | Func | | Test | |
|---|---|---|---|---|
| Human → | Yes | No | Yes | No |
| GPT-4o: Yes | 13 | 0 | 9 | 1 |
| GPT-4o: No | 1 | 6 | 4 | 6 |

Table 2: Agreement between human and GPT-4o on checking (1) the functionality equivalence between the sandboxed and original function, and (2) the test correctness. When GPT-4o predicts "Yes" for both quality checks, it has a high agreement with human.

| % Solved | % Use-Tool | Easy / Med / Hard |
|---|---|---|
| 81.5% | 59.3% | 29.6 / 51.9 / 18.5 |

Table 3: Human study results. We ask the participants to complete the function with the same context we use to evaluate code generation models in §5.

the train set and 100% for the evaluation set), we provide the LLM with the missing lines and prompt it to improve the test function by incorporating more tests.

**Final-Step Quality Check & Filtering**. As a final step, we conduct two quality checks: functionality equivalence check and test correctness check, and filter out unqualified examples.

To ensure the validity of sandbox testing, we examine the **functionality equivalence** between the sandboxed and original function. We first compare the AST of the function bodies, which is a sufficient condition of functionality equivalence. In our resulting dataset, 81.7% of the examples have the same AST. The remaining ones are filtered out from the evaluation set. To include more examples in the train set, since it is also possible that code with the same functionality has different ASTs (e.g., HTML tags can be parsed with `LexborHTMLParser` and `BeautifulSoup`), we prompt an LLM [2] to compare the functionality equivalence, which is shown to have high agreement with human (see our human study in §3.1 for details). We also include the examples that pass the LLM check for the train set.

In principle, a test that calls the target function without any assertion checks still has a 100% coverage rate. We hence conduct **test correctness** check and apply an LLM to check whether the tests are correct, reasonable, and are completing the verification of the functionality. Human study in §3.1 demonstrates the high precision of the LLM test correctness checker.

## 3 Resulting Datasets: REPOST-TRAIN and REPOST-EVAL

**Statistics**. We build a train set, REPOST-TRAIN, and an evaluation set, REPOST-EVAL. To reduce contamination, we build REPOST-TRAIN from repositories created between 2023-01-31 and 2024-08-31, and build REPOST-EVAL from repositories created after 2024-09-01.

We compare the statistics of our datasets and existing *execution-based* datasets in Table 1. To our knowledge, REPOST-TRAIN is currently the largest repo-level dataset with execution feedback. Prior works such as RepoEval (Zhang et al., 2023) or SWE-Gym (Pan et al., 2024) require huge human effort to set up the environments. Automated frameworks such as R2E (Jain et al., 2024b) apply LLMs to the complicated task of building the entire repositories and suffer from low success rates. In comparison, REPOST only requires

---

[2]The original code could be inexecutable due to package installation errors (e.g., `BeautifulSoup` in this case), so we do not execute the original code.

necessary dependencies for each function, which is more feasible for LLMs and benefits scalability.

Table 4 shows detailed statistics of our datasets. We achieve high test numbers and branch coverage rates with iterative test coverage improvement and quality filtering. Results also show that REPOST can create relatively complex examples in terms of length and local and external dependencies. The percentage of standalone functions (i.e., functions without local dependencies)

| Avg Stats. | TRAIN | EVAL |
|---|---|---|
| Target # Tokens (Lines) | 112.4 (12.8) | 102.7 (9.9) |
| Eval Script # Tokens (Lines) | 842.5 (75.7) | 1217.5 (122.3) |
| # Test Cases | 5.7 | 8.2 |
| Test Branch Coverage | 97.8% | 100% |
| % Standalone Functions | 28.1% | 26.4% |
| # External Libraries | 894 | 106 |

Table 4: Detailed statistics of our datasets.

are 28.1% and 26.4% in our datasets. Both are very close to 27%, the percentage of standalone functions among all GitHub code estimated by Li et al. (2024b).

## 3.1 Quality Verification with Human Study

**Agreement between LLM Checker and Human**. We conduct a human study to verify the precision of our LLM-based quality check strategies introduced in §2.4. We ask 3 computer science graduate students to conduct functionality equivalence and test correctness checks for 20 examples sampled from REPOST-EVAL (before the final filtering), with the same instruction we use to prompt the LLM checkers. The Kappa agreement scores among human annotators are 0.9179 for the functionality check and 0.7750 for the test check.

Results in Table 3 show that all 13/20 examples that pass GPT-4o's functionality check are also predicted as "same functionality" by human. Among 10 examples that pass GPT-4o's test quality check, 9 of them are predicted as "high-quality tests" by human. It demonstrates that after applying our filtering strategies for quality control, the remaining examples have high quality. In principle, one can further enhance dataset quality by manually inspecting and selecting the examples. We provide more details about the experiments in §A.2.

**Solvability Check**. We conduct another human study to check whether the examples are reasonable and can be solved by human. We assigned 27 examples constructed by REPOST to 9 computer science students, with no overlaps, and asked them to complete the function and answer questions about the difficulties of the examples (see §A.2 for details).

Results show that 81.5% of the examples were solved by human, indicating that most examples are reasonable and not too complicated. The remaining examples were not solved for various reasons, such as the participant is not familiar with the task (e.g., reading html data) or related libraries (e.g., BeautifulSoup4), the intent of the function cannot be fully entailed from the context, etc. Towards the unclear intent issue, we designed an experiment in §5, where we generated additional instructions to improve the clarity. We also observe that the generated examples have varying complexity levels based on the usage of external tools and the difficulty ratings. Furthermore, 33.3% of the examples are solved in the first submission and 37.0% require more than 5 submissions to solve.

## 4 Code Generation Training Experiments

**Training with REPOST-TRAIN**. In standard supervised fine-tuning (**SFT**), we can train the model with the code context $c$ as the input and the ground truth target function $f^*$ as the output. The execution feedback provided by our REPOST evaluation scripts further allows us to employ the **rejection sampling fine-tuning (RFT)** algorithm to generate additional valid training targets. Specifically, we apply the model itself to our dataset, generating $n$ candidate solutions for each function based on its code context: $(c, f_1), \ldots (c, f_n)$. The method is denoted as **RFT (Self)**. We can also apply other stronger models to generate candidates (denoted as **RFT (Distill)**). Only solutions that pass our test cases are retained. We then finetune the model on the successful functions $(c, f'_1), \ldots (c, f'_m)$ and the ground truth $(c, f^*)$ using the standard negative log-likelihood loss.

| Model | HumanEval | | RepoEval-Func | | REPOST-EVAL | |
|---|---|---|---|---|---|---|
| | Pass@1 | Δ | Pass@1 | Δ | Pass@1 | Δ |
| StarCoder2-7B (Lozhkov et al., 2024) | 34.76 | – | 32.98 | – | 26.35 | – |
| + SFT | 37.20 | ↑2.44 | 33.78 | ↑0.80 | 27.70 | ↑1.35 |
| + RFT (Self) | 39.63 | ↑4.87 | 34.58 | ↑1.61 | 28.38 | ↑2.03 |
| + RFT (Distill) | **40.24** | ↑**5.49** | **35.12** | ↑**2.14** | **29.05** | ↑**2.70** |
| Qwen2.5-Coder-7B (Hui et al., 2024) | 79.27 | – | 38.06 | – | 29.39 | – |
| + SFT | 80.48 | ↑1.21 | 39.94 | ↑1.88 | 30.74 | ↑1.35 |
| + RFT (Self) | **84.76** | ↑**5.49** | 40.75 | ↑2.69 | 31.76 | ↑2.36 |
| + RFT (Distill) | **84.76** | ↑**5.49** | **41.55** | ↑**3.49** | **32.43** | ↑**3.04** |

Table 5: Code generation training results. We evaluate Pass@1 for all experiments. For RepoEval, we use the "Oracle" repo-level context as used in their original paper.

To obtain more training pairs for both the "Self" and "Distill" settings, we further prompt the model itself or a stronger model to debug the failed solutions, with the error message in the context. We also train the model on successfully debugged solutions $(c, f_1''), \ldots (c, f_k'')$.

### 4.1 Experimental Setup

**Datasets and Evaluation Metrics**. We train two models: StarCoder2-7B (Lozhkov et al., 2024) and Qwen2.5-Coder-7B (Hui et al., 2024) with REPOST-TRAIN and evaluate on two public benchmarks: HumanEval (Chen et al., 2021), an algorithm problem dataset, and RepoEval (Zhang et al., 2023), a repo-level code generation dataset. We also evaluate on REPOST-EVAL. For RepoEval, we only evaluate on the "function" split, which supports execution. We use the "Oracle" context to mitigate the bias of context retrieval methods. We report the Pass@1 scores on all datasets. More evaluation details are provided in §A.3.

**Training Details**. We compare three training methods: SFT, RFT (Self), and RFT (Distill). For the "Distill" method, we apply GPT-4o and Claude-3.5-Sonnet to generate and debug candidate solutions, separately, and combine their successful candidates for training. We provide the number of examples where we obtained at least one successful solution in Table 9. Other details are shown in subsection A.3.

### 4.2 Experimental Results

**Main Results**. Table 5 demonstrates that models trained with REPOST-TRAIN can generalize well to other public benchmarks. Specifically, we improve Qwen2.5-Coder by 5.5% Pass@1 on HumanEval, 3.5% on RepoEval-Func, and 3.0% on REPOST-EVAL. Furthermore, in all experiments, training with RFT, even with self-training only, achieves better performance than finetuning on the original GitHub function only. For instance, RFT (Distill) outperforms SFT by 4.3% Pass@1 on HumanEval. This shows the benefit of training with environments that can provide execution feedback. We can also observe that RFT with self-training in general has lower performance than distilling from stronger models. As shown in Table 9, we can only obtain 1573 additional training targets with StarCoder2, but we obtained 3606 by combining GPT-4o and Claude-3.5. We hypothesize that one can obtain more examples and hence achieve better performance by sampling with more candidate solutions, generating from different types of context, etc., and we leave that to future work.

**Scaling Law Analysis**. In Figure 4a, we investigate how the scale of training data affects model performance. Specifically, we randomly sample different numbers of examples from REPOST-TRAIN to train the model. We can see that the performance of RFT (Distill) increases as we scale up the training data, which suggests the advantage of training data with high scalability. Furthermore, with different scales of data, RFT consistently achieves better performance than SFT, which further demonstrates the effectiveness of training environments with execution feedback.

| Model | Pass@1 |
|---|---|
| CodeLlama-7B (Rozière et al., 2024) | 25.68 |
| StarCoder2-7B (Lozhkov et al., 2024) | 26.35 |
| Qwen2.5-Coder-7B (Hui et al., 2024) | 29.39 |
| MagiCoder-S-DS-6.7B (Wei et al., 2023) | 33.78 |
| CodeLlama-34B (Rozière et al., 2024) | 32.43 |
| Qwen2.5-Coder-32B (Hui et al., 2024) | 33.11 |
| DS-R1-Qwen-32B (DeepSeek-AI, 2025) | 34.46 |
| CodeLlama-70B (Rozière et al., 2024) | 32.43 |
| DS-R1-Llama-70B (DeepSeek-AI, 2025) | 33.45 |
| GPT-4o-mini (OpenAI, 2024) | 35.81 |
| Claude-3.5-Sonnet (Anthropic, 2024) | 37.16 |
| GPT-4o (OpenAI, 2023) | **39.53** |

Table 6: Code generation results on REPOST-EVAL. The model/open-source model with the best performance is highlighted in **bold**/underlined.

| Repair Iter | 0 | 1 | 2 | 3 |
|---|---|---|---|---|
| GPT-4o-mini | 35.81 | 43.58 ↑7.77 | 46.62 ↑3.04 | **47.97** ↑1.35 |
| GPT-4o | 39.53 | 48.31 ↑8.78 | 52.36 ↑4.05 | **53.72** ↑1.35 |

Table 7: Performance on REPOST-EVAL with self-repairing. We show the Pass@1 gain compared to the previous repairing iteration.

| Docstring (DocS) | w/o DocS | w/ Orig DocS | Gen-DocS |
|---|---|---|---|
| GPT-4o-mini | 37.28 | 38.98 | **55.93** |
| GPT-4o | 39.55 | 42.37 | **61.02** |

Table 8: Performance on the subset of REPOST-EVAL examples. The performances of both models are largely improved with model-generated docstrings.

**Repository Diversity Experiment**. Figure 4b examines whether broader repository coverage leads to better performance, given a fixed budget of training examples. We fix the number of examples to 2,000 and experiment with two example sampling methods: (1) Sample-by-Repo, where we keep sampling repositories and adding all the functions in the repository to the training set, until the data size reaches 2,000; (2) Sample-by-Example, which is the same setting as Figure 4a, where we randomly sample functions from REPOST-TRAIN. We observe that Sample-by-Example, covering 678 repositories, outperforms Sample-by-Repo, which only covers 221. The performance is further improved by RFT. Recall that our method only needs to set up individual functions, compared to existing methods, such as R2E, that need to build the entire repository, it is much easier to build datasets with broad repository coverage with our method, which benefits model training.

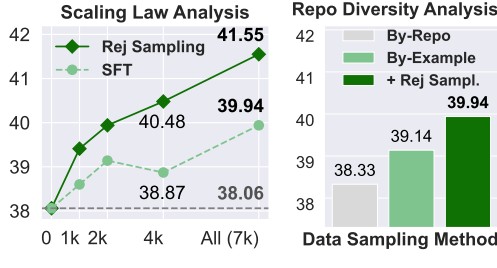

(a) Scaling law analysis.     (b) Repo diversity.

Figure 4: (a) Pass@1 scores on RepoEval with different numbers of training examples. (b) Pass@1 scores on RepoEval with different methods to sample 2,000 training examples. Sample-by-Example has a broader repository coverage and achieves better Pass@1. The performance is further enhanced with RFT (Distill).

## 5 Benchmarking with REPOST-EVAL

**Experimental Setup**. We benchmark LLMs on REPOST-EVAL to evaluate their abilities to generate code in real GitHub repositories. Specifically, we prompt the LLMs to generate the target functions based on code context retrieved from the repository. To ensure that we cover all the relevant dependencies in the context, we follow R2E (Jain et al., 2024b) and build the context by extracting modules that the target function depends on. Then we copy model solutions to the evaluation scripts for execution and compute the Pass@1 scores (Chen et al., 2021): the fraction of generated solutions that pass all test cases.

**Main Results**. Table 6 presents the code generation results of open-source and proprietary models. The best model (GPT-4o) only achieves 39.53 Pass@1, which shows that our benchmark is challenging and has a large room for improvement. There is still a gap of 5.07 Pass@1 between the best open-source (DS-R1-Qwen-32B) and proprietary model (GPT-4o), which calls for future work to improve open-source models by training.

**Self-Repairing Experiments**. In addition to static code generation, we also study the effect of self-repairing (Chen et al., 2024), which allows the models to iteratively repair solutions based on the error message and stack trace and has been shown to be effective for code generation. As shown in Table 7, both models consistently benefit from self-repairing, but the performance gain becomes saturated after 3 debugging iterations. We can also see that the performance gap between the two models becomes larger when we allow more iterations, indicating that GPT-4o is stronger in both code generation and debugging.

**Generation with Model-Generated Docstrings**. In our human study (§3.1), we observe that in some examples, the intent of the target function cannot be fully inferred from the context. The same issue is reported in existing repo-level benchmarks such as R2E (Jain et al., 2024b). In this experiment, we investigate whether model-generated docstrings can provide better specifications. On the subset of REPOST-EVAL examples that have docstrings, we evaluate code generation (1) with the docstring removed, (2) with the original docstring, and (3) with GPT-4o-generated docstrings, with the function and its dependencies as the context.

We observe that both models achieve much higher performance when provided with generated docstrings. One possible explanation is that the quality of human-written docstrings varies widely. Some docstrings may only contain limited information, while the generated docstrings generally contain more details. On average, the original docstrings have 36.82/3.95 tokens/lines, while the GPT-4o-generated docstrings have 130.84/9.92. Furthermore, with more detailed docstrings, the performance gap between the two models becomes larger. When the specifications are not clear, it could be impossible to complete the function as intended, causing both weak and strong models to fail.

# 6    Related Work

**Code Generation Training**.  Existing works have shown the effectiveness of pretraining (Rozière et al., 2024; Lozhkov et al., 2024; Guo et al., 2024) or instruction tuning (Wei et al., 2023; Luo et al., 2023; DeepSeek-AI, 2025) on real-world code. To further finetune models for code generation, existing works have built large-scale training datasets with test cases by leveraging large-scale online algorithm problems. The execution feedback from test cases is used in constructing training targets (Ni et al., 2024; Zhang et al., 2024) or reward signals (Liu et al., 2023a; Jiang et al., 2024). As repo-level code generation, existing works such as SWE-Gym (Pan et al., 2024) build training environments by manually setting up dependencies and configurations for the entire GitHub repositories. The repository setup process is complicated and challenging to automate, resulting in limited dataset scales. In comparison, we design a sandbox testing method that only requires setting up the necessary dependencies for individual GitHub functions, which reduces the difficulty of environment setup and leads to better scalability.

**Code Generation Benchmarks**. Execution-based benchmarks have been widely adopted for code generation, which provide test cases to evaluate the generated code (Chen et al., 2021; Hendrycks et al., 2021; Austin et al., 2021). Researchers have built benchmarks with large test coverage (Liu et al., 2023b), multiple languages (Cassano et al., 2022), diverse domains (Lai et al., 2023; Du et al., 2023), etc. LiveCodeBench (Jain et al., 2024a) periodically extracts newly released algorithm problems, which enables contamination-free evaluation.

To assess the models' ability on repo-level coding tasks, recent works leverage repositories with test cases to build benchmarks on code patch generation (Zhang et al., 2023; Xie et al., 2024), issue solving (Jimenez et al., 2024), test generation (Jain et al., 2025), test execution (Bouzenia & Pradel, 2024), environment setup (Bogin et al., 2024), etc. Recent works aim to provide live evaluation manually (Li et al., 2024a) or leverage an LLM to set up the repository environment and generate test cases. However, both methods require building the entire repository, which is challenging for both human and LLMs and limit the scale of the resulting datasets. REPOST improves the scalability with sandbox testing and enables the construction of live benchmarks from naturally occurring repositories.

# 7 Conclusion and Future Works

We present REPOST, a scalable method to construct environments for code generation in real-world repositories that support sandbox testing. REPOST is fully automatic and enables the construction of scalable execution-based training environments and live benchmarks. Experiments demonstrate that training with the resulting train set, REPOST-TRAIN, leads to performance gain on other public benchmarks. For instance, we improve Qwen2.5Coder by 5.49%/3.49% Pass@1 on HumanEval/RepoEval. We also build an evaluation set, REPOST-EVAL, to showcase the potential of REPOST to construct live benchmarks.

**Future works** may include: (1) further scaling up the datasets with more input repositories, (2) exploring the utility of different types of context in training and evaluation, (3) adapting REPOST to other repo-level coding tasks such as issue-solving (Pan et al., 2024), code translation (Xie et al., 2023), code refactoring (Gautam et al., 2025), environment setup (Bogin et al., 2024), etc., and (4) using REPOST-TRAIN to train and evaluate coding agents (Yang et al., 2024; Wang et al., 2025). This is possible because our datasets provide both access to the original GitHub repositories and execution feedback. Specifically, one can set the instruction as "generate the target function", and the coding agent will need to explore and interact with the entire repository by itself to obtain relevant information. We can then use the evaluation scripts to select successful trajectories and use them for model training.

## Reproducibility Statement

We provide the following ways to reproduce our results: (1) We release the code for the entire REPOST pipeline, including repository and function curation, sandboxing, test generation, execution, and final-stage quality verification. (2) We release the code for training data construction, including both the SFT and RFT settings. (3) We release the REPOST-TRAIN and REPOST-EVAL datasets, including the repository commit ids and the evaluation scripts generated by REPOST. We also release the RFT (Distill) data constructed based on REPOST-TRAIN. (4) We release the docker images of REPOST-TRAIN and REPOST-EVAL, which already install all the external packages required for executing all the evaluation scripts.

All the above resources can be found at `https://repost-code-gen.github.io/`.

# 8 Acknowledgement

We thank Ofir Press, Saujas Vaduguru, Atharva Naik, Yuning Mao, and Xuhui Zhou for their helpful feedback on this work. We thank all participants in our human study. This work was supported in part by NSF grant DSES 2222762. Yiqing Xie is supported by the Carnegie Mellon University Presidential Fellowship in the Language Technologies Institute.

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

# A  Appendix

## A.1  Data Construction Details

**Sandboxing and Test Generation Details**. Table 14 shows the prompt we use to sandbox the target function and its local dependencies to a separate evaluation script. Table 15 shows the prompt template we use to generate tests in the evaluation scripts.

**Quality Control Details**. Table 16 shows the prompt we use to debug the evaluation scripts if there are any errors when we copy the ground truth solution as the new implementation and execute the scripts. Table 17 shows the prompt we use to improve the coverage of the test function if there are any missing branches. Table 18 shows the prompt we use in the quality check stage, where we check whether the sandboxed and original functions have exactly the same functionality. Table 19 shows the prompt we use in the quality check stage, where we check whether the generated test function is correct, reasonable, and are completing the verification of the functionalities of the ground truth function and the new implementation.

## A.2  Human Study Details

**Quality Check Agreement Check**. For functionality equivalence check, we randomly sample 20 examples from REPOST-EVAL. The instruction we present to the participants is exactly the same as the prompt for the LLM functionality checker, as shown in Table 18. Specifically, we show them the original and sandboxed functions and ask them whether their functionalities are the same. We allow minor differences including additional sanity checks or different print information.

As for the test correctness check, we randomly select 10 examples where the LLM predicts as "Yes" and 10 examples where the LLM predicts as "No". The instruction is the same as the prompt for the LLM test correctness check, as shown in Table 19. We additionally prevent checking the values of printed or logged information because it is typically difficult to match the exact same information in code generation.

**Solvability Check**. Similar to the setting we use to benchmark coding models in §6, we show the participants the direct or indirect dependencies of the target function and ask them to complete it. After submitting an answer, the participants will see the execution results of our evaluation scripts and can choose to revise their answers accordingly or to give up. Finally, we asked them whether they used external tools (e.g., search engines) in completing the function and asked them to rate the difficulty of each example. Note that we do not directly show the evaluation script to the participants and explicitly ask them not to use any AI models.

## A.3  Code Generation Training Details

**Evaluation Details**. The performance of repo-level code generation also depends on the quality of the retrieved context. To mitigate the bias of retrieval models, we evaluate the models with the "Oracle" context for the two repo-level datasets. For RepoEval-Func, we follow the setting in their original paper that retrieves in-repo code fragments with the target function as the query. For REPOST-EVAL, to ensure that we cover all the relevant dependencies in the context, we follow R2E (Jain et al., 2024b) and use dependency-only context, which includes modules that the target function depends on with the call graph.

**Training Details**. We train the models with a learning rate of $2e-6$, a batch size of 32, and a warm-up ratio of 0.1 for 1 epoch. For QwenCoder, we add a linebreak after the prompt to prevent the first token of the target output from being linebreak.

**Model Behavior Analysis**. Table 10 shows that the model learns to generate more comments after RFT compared to the base model or SFT. An explanation is that when generating candidate implementations, GPT-4o and Claude-3.5 generate more comments than the

| Docstring (DocS) | Initial Generation | w/ Debugging |
|---|---|---|
| StarCoder2-7B | 1327 / 7415 | 1573 / 7415 |
| Qwen2.5-Coder-7B | 1428 / 7415 | 1708 / 7415 |
| GPT-4o | 2438 / 7415 | 2861 / 7415 |
| Claude-3.5 | 2503 / 7415 | 3092 / 7415 |
| GPT-4o & Claude-3.5 | **3110** / 7415 | **3606** / 7415 |

Table 9: The number of examples where we obtained at least one successful solution in RFT (Self) and RFT (Distill). We report the number (1) after the initial round of generation, and (2) after debugging.

| Method | Avg # of Comments | |
|---|---|---|
| | Train Data | Output on RepoEval |
| Qwen2.5-Coder-7B | – | 0.48 |
| + SFT | 0.69 | 0.56 |
| + RFT (Distill) | 1.06 | 0.61 |

Table 10: Number of comments in (1) the training targets and (2) generations on the REPOST-EVAL dataset.

---

***Sanity Checks for Sandboxing***

1. The target function should exist in the evaluation script.
2. The number of tokens in the sandboxed function should NOT be more than 20 fewer than that in the original function.
3. The number of tokens in the entire evaluation script should NOT be more than 50 fewer than the number of tokens in the combination of all local dependencies.

---

Table 11: The sanity checks we conduct for the sandboxing step.

human-written function. This indicates that training with RFT enables the model to output more interpretable and well-documented code.

## A.4 Error Analysis on RepoEval

As shown in Table 13, we randomly sampled 10 failure examples from RepoEval and manually checked the causes of error.

Among the 10 failure cases, 6/10 have unclear instructions or unclear package versions, which means they cannot even be solved by human. This is a known issue with the RepoEval dataset, where the intention of the functions cannot be fully entailed from the context.

Since performance is calculated over the total set of examples, including many that are inherently unsolvable or ambiguous, we believe adjusting for this by considering only clearly solvable cases would highlight a more substantial gain from our method.

We observe similar issues with our RepoST-Eval datasets and present a potential solution: using model-generated docstrings to improve clarity (see section 5 for details).

| *Sanity Checks for Test Generation* |
| --- |
| 1. The target function should exist in the evaluation script. |
| 2. The number of tokens in the sandboxed function should NOT be more than 20 fewer than that in the original function. |
| 3. The number of tokens in the entire evaluation script should NOT be fewer than the number of tokens in the evaluation script obtained from the sandboxing step. |
| 4. A test function named test_{func_name}() should exist. |
| 5. The test function should call the target function. |
| 6. The test function should call the new implementation (new_implementation_{func_name}()). |
| 7. There should be at least 3 assertions in the test function. |
| 8. There should be a main function. |
| 9. The main function should call the test function. |
| 10. If the main function calls the test function, the function call should not be in a try-except block. |

Table 12: The sanity checks we conduct for the test generation step.

| Issue | Frequency |
| --- | --- |
| Instruction not clear | 2/10 |
| Package version mismatch (e.g., transformers) | 1/10 |
| Problem with boundary cases | 4/10 |
| Correct functionality but wrong implementation | 1/10 |
| Wrong functionality | 1/10 |
| Format error (e.g., does not generate a function) | 1/10 |

Table 13: Summary of common error types on RepoEval. The examples are randomly sampled.

**Sandboxing Prompt**

Instructions:
- You're given a piece of PYTHON CODE containing a function called {func_name}. We also provide you the CONTEXT of the PYTHON CODE. Your goal is to aggregate the PYTHON CODE and the CONTEXT into one script, so that we can directly call the {func_name} function WITHOUT ANY MODIFICATIONS.
- You should edit the original PYTHON CODE as little as possible and you can add code only if necessary.
- DO NOT call any external API, database, etc. Instead, create a mock interface.
- Make sure that your code can be directly executed without any modification. For example, statements like 'token = "your_auth_token_here" # You need to replace this with a real token' is NOT allowed.
- If you need to write files to the disk, use '{docker_CACHE_DIR}' as the directory.

- Provide your reasoning and the revised PYTHON CODE below SOLUTION.

PYTHON CODE:
"'python
{code}
"'

CONTEXT:
{context}

Your answer should follow the format below:

Reasoning: ...
"'python
# Your Code.
"'

Do NOT include other formatting. Output every token of the content with no omission or abbreviation. For example, abbreviation like '... # the code keeps unchanged' is NOT allowed.

SOLUTION:

Table 14: The prompt we use to sandbox the target function and its local dependencies to a separate evaluation script.

**Test Generation Prompt**

Instructions:
- You're given a piece of PYTHON CODE containing a function called {func_name}. Assume we will later have another implentation of the {func_name} function called {func_name}_new_implementation.
- Your goal is to add (1) a test function called {test_func_name} to check whether {func_name}_new_implementation has the same functionality as the {func_name} function, and (2) a __main__ function that calls the test function.
- If the PYTHON CODE already contains a __main__ function, remove it and write a new __main__ function.
- The test function {test_func_name} should contain at least 3 assert statements. If {func_name}_new_implementation has different functionality as {func_name}, an Assertion Error should be triggered.
- The test function {test_func_name} should cover all the major branches of the {func_name} function
- DO NOT test on error handling and DO NOT test on the print information in the function.
- The __main__ function should NOT contain a try-except structure. If the implementation is incorrect, the program should have a non-zero exit code.
- You should edit the original PYTHON CODE as little as possible.
- If you need to write files to the disk, use '{docker_CACHE_DIR}' as the directory.

- Provide your reasoning and the new PYTHON CODE containing your test function {test_func_name} and the __main__ function below SOLUTION.

PYTHON CODE:
"'python
{code}
"'

Your answer should follow the format below:

Reasoning: ...
"'python
# The new PYTHON CODE containing your test function {test_func_name} and the __main__ function.
"'

Do NOT include other formatting. Output every token of your edited PYTHON CODE with no omission or abbreviation.

SOLUTION:

Table 15: The prompt we use to generate tests in the evaluation scripts.

---

*Debugging Prompt* (for data construction)

Instructions:
- You're given a piece of PYTHON CODE containing a function called {func_name} and its test function called {test_func_name}. Assume we will later add another function called {func_name}_new_implementation, the test function aims to check whether {func_name}_new_implementation has the same functionality as {func_name}.
- In our experiments, we implemented {func_name}_new_implementation exactly the same as {func_name}, but the PYTHON CODE cannot be successfully executed.
- Your task is to debug PYTHON CODE based on the ERROR MESSAGE.
- You should modify the code as little as possible, especially the test_{func_name} function and the {func_name} function.
- Make sure that after debugging, the test function test_{func_name} still have at least three assert statements and cover all the major branches of the {func_name} function.
- DO NOT test the logging information of error handling and DO NOT test on the print information in the function.
- If you need to write files to the disk, use '{docker_CACHE_DIR}' as the directory.

- Provide your reasoning and the debugged PYTHON CODE below SOLUTION. If necessary, output the bash scripts for Linux in another code block in the format of "'bash ... "'.

PYTHON CODE:
"'python
{code}
"'

ERROR MESSAGE:
"'
{err_msg}
"'

Your answer should follow the format below:

Reasoning: ...
"'python
# The debugged PYTHON CODE in one piece.
"'

"'bash
# the bash script, if necessary
"'

Do NOT include other formatting. Output every token of your debugged PYTHON CODE with no omission or abbreviation.

SOLUTION:

---

Table 16: The prompt we use to debug the evaluation scripts if there are any errors when we copy the ground truth solution as the new implementation and execute the scripts.

---

*Test Coverage Improvement Prompt*

Instructions:
- You're given a piece of PYTHON CODE containing a function called {func_name} and its test function called {test_func_name}. Assume we will later add another function called {func_name}_new_implementation, the test function aims to check whether {func_name}_new_implementation has the same functionality as {func_name}.
- You're also given the MISSING LINES of the {func_name}_new_implementation function that are NOT covered by {test_func_name}.
- Your task is to improve the branch coverage rate of the {test_func_name} function.
- You should only modify the {test_func_name} function. DO NOT modify other parts of the code.
- DO NOT test the logging information of error handling and DO NOT test on the print information in the function.
- If you need to write files to the disk, use '{docker_CACHE_DIR}' as the directory.

- Provide your reasoning and your revised {test_func_name} function below SOLUTION.

PYTHON CODE:
"'python
{code}
"'

MISSING LINES:
{missing_code}

Your answer should follow the format below:

Reasoning: ...
"'python
# Your revised {test_func_name} function
"'

Do NOT include other formatting. Output every token of the {test_func_name} function with no omission or abbreviation.

SOLUTION:

---

Table 17: The prompt we use to improve the coverage of the test function if there are any missing branches.

---

*Functionality Equivalence Check Prompt*

---

Instructions:
- We revised a python function called {func_name} so it can be directly executed in an isolated environment.
- You are given the ORIGINAL FUNCTION and the CODE containing the REVISED FUNCTION.
- Your task is to check whether the functionality of the REVISED FUNCTION is the same as the ORIGINAL FUNCTION.
- If the REVISED FUNCTION is exactly the same as the ORINIGAL FUNCTION, output "same" as your answer.
- Otherwise, if the functionality of the REVISED FUNCTION is the same as the ORIGINAL FUNCTION, output "yes" as your answer.
- if the functionality of the REVISED FUNCTION is different, output "no".

- Provide your reasoning and the answer under "SOLUTION".

ORIGINAL FUNCTION:
{orig_func}

CODE containing the REVISED FUNCTION:
{new_code}

Your answer should follow the format below:
"'
REASONING: Your reasoning,
ANSWER: "same", "yes" or "no".
"'

Do NOT include other formatting.

SOLUTION:

---

Table 18: The prompt we use in the quality check stage, where we check whether the sandboxed and original functions have exactly the same functionality.

---

*Test Correctness Check Prompt*

---

Instructions:
- You are given a piece of PYTHON CODE containing a function called {func_name}, its new implementation {func_name}_new_implementation (now hidden) and its test function called {test_func_name}.
- Your task is to judge whether the test function satisfies all the CONDITIONS:
** CONDITION 1 ** The {func_name} function should either have return values or modifies global variables or input arguments (such as a list, a dictionary, a class, etc.).
** CONDITION 2 ** The test cases should only check the return values or variable states. It should NOT check printed or logged contents.
** CONDITION 3 ** {func_name}_new_implementation can pass all the test cases IF AND ONLY IF it has the EXACTLY same functionality as {func_name}.
** CONDITION 4 ** The test cases and assert statements are reasonable. For example, if {func_name} does not have return values, you should NOT use 'assert {func_name}() == {func_name}_new_implementation()' to test the implementation.
** CONDITION 5 ** The test cases are non-trivial.

- If the test function satisfies all the CONDITIONS, answer "yes". Otherwise, answer "no".
- Provide your reasoning and the answer under "SOLUTION".

PYTHON CODE:
{code}

Your answer should follow the format below:
"'
REASONING: Your reasoning,
ANSWER: "yes" or "no".
"'

Do NOT include other formatting.

SOLUTION:

---

Table 19: The prompt we use in the quality check stage, where we check whether the generated test function is correct, reasonable, and actually comparing the functionalities of the ground truth function and the new implementation.

**[Case 1, Origifinal Function & GitHub Context]**  `score_explicit_question()`

```python
## chan_questions.py
import json
import prompts

from helpers import get_openai_answer, chunker, clean_and_hash, clean_html, query_to_search_url

def score_explicit_question(string: str) -> list:
    """
    Uses LLMs to score a question based on whether it is considered explicit or implicit.
    Uses OpenAI.
    """
    prompt = prompts.IS_EXPLICIT
    answer = get_openai_answer(prompt.replace("[input]", string))

    results = json.loads(answer)["results"]
    return results
```

```python
## helpers.py
import openai
import config

def get_openai_answer(prompt: str, response_format="json_object", model=None):
    # initiate
    client = openai.OpenAI(api_key=config.OPENAI_KEY)

    if not model:
        model = config.MODEL

    # Get response
    response = client.chat.completions.create(
        model=model,
        temperature=config.TEMPERATURE,
        max_tokens=config.MAX_OUTPUT_TOKENS,
        response_format={"type": response_format},
        messages=[{
            "role": "user",
            "content": prompt
        }]
    )

    return response.choices[0].message.content
```

Figure 5: Case study 1. The original score_explicit_question function and its context extracted from the original GitHub repository. The function calls the text completion function from the OpenAI API.

```
[Case 1, Evaluation Script - Sandboxing]   score_explicit_question()

import json

# Mock configuration
class config:
    OPENAI_KEY = "mock_openai_key"
    MODEL = "mock_model"
    TEMPERATURE = 0.5
    MAX_OUTPUT_TOKENS = 100

# Mock prompts
class prompts:
    IS_EXPLICIT = "Is the following question explicit or implicit? [input]"

# Mock OpenAI API response
def get_openai_answer(prompt: str, response_format="json_object", model=None):
    # Mock response simulating the OpenAI API
    mock_response = {
        "results": [
            {"question": prompt, "score": "explicit" if "explicit" in prompt else "implicit"}
        ]
    }
    return json.dumps(mock_response)

# The original function from chan_questions.py
def score_explicit_question(string: str) -> list:
    """
    Uses LLMs to score a question based on whether it is considered explicit or implicit.
    Uses OpenAI.
    """
    prompt = prompts.IS_EXPLICIT
    answer = get_openai_answer(prompt.replace("[input]", string))

    results = json.loads(answer)["results"]
    return results
```

Figure 6: Case study 1. The sandboxed score_explicit_question function in the evaluation script. The LLM generates a mock function called get_openai_answer to replace the real API call. With the mock class, the score_explicit_question has the exactly same functionality as the original function, but does not make real OpenAI API calls.

---

**[Case 1, Evaluation Script - Tests]**    `score_explicit_question()`

```python
def test_score_explicit_question():
    # Test case 1: Explicit question
    input_str1 = "Is this an explicit question?"
    assert score_explicit_question(input_str1) == score_explicit_question_new_implementation(input_str1)

    # Test case 2: Implicit question
    input_str2 = "What is the meaning of life?"
    assert score_explicit_question(input_str2) == score_explicit_question_new_implementation(input_str2)

    # Test case 3: Neutral question (contains neither explicit nor implicit)
    input_str3 = "How are you?"
    assert score_explicit_question(input_str3) == score_explicit_question_new_implementation(input_str3)

    # Test case 4: Very short question
    input_str4 = "?"
    assert score_explicit_question(input_str4) == score_explicit_question_new_implementation(input_str4)

    # Test case 5: Very long question
    input_str5 = "Is this an explicit question?" * 100
    assert score_explicit_question(input_str5) == score_explicit_question_new_implementation(input_str5)

    # Test case 6: Case sensitivity
    input_str6 = "is this an explicit question?"
    assert score_explicit_question(input_str6) == score_explicit_question_new_implementation(input_str6)

    # Test case 7: Different punctuation
    input_str7 = "Is this an explicit question!"
    assert score_explicit_question(input_str7) == score_explicit_question_new_implementation(input_str7)

    # Test case 8: Special characters
    input_str8 = "Is this an explicit question? #$%"
    assert score_explicit_question(input_str8) == score_explicit_question_new_implementation(input_str8)

    # Test case 9: Numbers in question
    input_str9 = "Is 42 an explicit number?"
    assert score_explicit_question(input_str9) == score_explicit_question_new_implementation(input_str9)

    # Test case 10: Empty string
    input_str10 = ""
    assert score_explicit_question(input_str10) == score_explicit_question_new_implementation(input_str10)

    # Test case 11: Leading and trailing whitespace
    input_str11 = "   Is this an explicit question?   "
    assert score_explicit_question(input_str11) == score_explicit_question_new_implementation(input_str11)

    # Test case 12: Excessive internal whitespace
    input_str12 = "Is   this    an    explicit    question?"
    assert score_explicit_question(input_str12) == score_explicit_question_new_implementation(input_str12)

# Main function
if __name__ == "__main__":
    test_score_explicit_question()
```

Figure 7: Case study 1. The tests in the evaluation scripts.

**[Case 2, Origifinal Function & GitHub Context]    `QuizBrain.next_question()`**

```python
## Complete Python Bootcamp/Day 017 Quiz Game/quiz_brain.py
class QuizBrain:
    def __init__(self, questions):
        self.question_number = 0
        self.question_list = questions
        self.score = 0

    def still_has_question(self):
        return self.question_number < len(self.question_list)

    def next_question(self):
        user_answer = ""
        current_question = self.question_list[self.question_number]
        while user_answer.lower() not in ("true", "false"):
            user_answer = ""
            user_answer = input(f"Q{self.question_number + 1}: {current_question.question} (True/False)")

        self.question_number += 1
        self.check_answer(user_answer, current_question.answer)

    def check_answer(self, user_answer, correct_answer):
        if user_answer.lower() == correct_answer.lower():
            print("Correct!")
            self.score += 1
        else:
            print("Incorrect")
        print(f"The correct answer is: {correct_answer}\nYour current score is {self.score}/{self.question_number}")
```

Figure 8: Case study 2. The original `next_question` function. The function reads from system inputs, which are not available when testing in a docker.

**[Case 2, Evaluation Script - Sandboxing]    `QuizBrain.next_question()`**

```python
class MockInput:
    def __init__(self, responses):
        self.responses = responses
        self.index = 0

    def __call__(self, prompt):
        if self.index < len(self.responses):
            response = self.responses[self.index]
            self.index += 1
            print(prompt + response)  # Simulate user seeing the prompt and typing the response
            return response
        raise Exception("No more mock inputs available.")

class Question:
    def __init__(self, question, answer):
        self.question = question
        self.answer = answer

class QuizBrain:
    def __init__(self, questions):
        self.question_number = 0
        self.question_list = questions
        self.score = 0

    def still_has_question(self):
        return self.question_number < len(self.question_list)

    def next_question(self):
        user_answer = ""
        current_question = self.question_list[self.question_number]
        while user_answer.lower() not in ("true", "false"):
            user_answer = ""
            user_answer = input(f"Q{self.question_number + 1}: {current_question.question} (True/False)")

        self.question_number += 1
        self.check_answer(user_answer, current_question.answer)

    def check_answer(self, user_answer, correct_answer):
        if user_answer.lower() == correct_answer.lower():
            print("Correct!")
            self.score += 1
        else:
            print("Incorrect")
        print(f"The correct answer is: {correct_answer}\nYour current score is {self.score}/{self.question_number}")
```

Figure 9: Case study 2. The sandboxed next_question function in the evaluation script. The LLM generates a mock class called `MockInput` to replace the real system input. With the mock class, the next_question has the exactly same functionality as the original function, but does not read system inputs.

```
[Case 2, Evaluation Script - Tests]   QuizBrain.next_question()

def test_next_question():
    questions = [
        Question("Is the sky blue?", "True"),
        Question("Is the grass red?", "False"),
    ]
    quiz_original = QuizBrain(questions)
    quiz_new = QuizBrain(questions)

    mock_responses = ["True", "False"]
    mock_input = MockInput(mock_responses)

    # Replace the built-in input function with mock_input for testing
    global input
    original_input = input
    input = mock_input

    # Run original implementation
    while quiz_original.still_has_question():
        quiz_original.next_question()

    # Reset input for the new implementation
    mock_input = MockInput(mock_responses)
    input = mock_input

    # Run new implementation
    while quiz_new.still_has_question():
        quiz_new.next_question_new_implementation()

    assert quiz_original.score == quiz_new.score, "Scores differ between implementations"
    assert quiz_original.question_number == quiz_new.question_number, "Question numbers differ between
implementations"
    assert quiz_original.still_has_question() == quiz_new.still_has_question(), "Question completion state differs
between implementations"

if __name__ == "__main__":
    test_next_question()
```

Figure 10: Case study 2. The tests in the evaluation scripts, which call the MockInput class to mock the system input.

