# OpenReview forum: "RepoST: Scalable Repository-Level Coding Environment Construction with Sandbox Testing"
_colmweb.org/COLM/2025/Conference — COLM 2025_

### Official Review · Reviewer_dCkp · 2025-05-12

**Rating:** 6
**Confidence:** 4
**Ethics Flag:** 1

**Summary:**

The paper addresses the challenge of constructing large-scale, executable repository-level coding datasets for code generation and evaluation. Existing approaches rely on full repository setups for integration testing, which are complex and error-prone. RepoST introduces a scalable alternative by isolating individual functions and their dependencies in sandboxed environments, enabling targeted execution feedback without extensive repository setup.

The RepoST framework comprises four main steps: repository and function curation, sandboxing with mock APIs for external dependencies, synthetic test generation, and rigorous quality control through iterative debugging and coverage checks.

The paper also introduces **RepoST-Eval**, a benchmark containing 296 functions from 99 repositories. The best model (GPT-4o) achieves only 39.53 Pass\@1, underscoring the benchmark’s complexity and potential for further research.

Additionally, the authors leverage **Rejection Sampling Fine-Tuning (RFT)** using execution feedback from RepoST, achieving notable improvements in Pass\@1 scores across benchmarks, including a 5.5% gain on HumanEval and a 3.5% gain on RepoEval for the Qwen2.5-Coder model.

**Reasons To Accept:**

**Quality and Clarity:**

* The paper is well-structured and clearly written, effectively presenting its motivation, dataset construction pipeline, and methodology with adequate visual aids and illustrative examples.
* The explanations of the proposed RepoST framework, from sandboxing to test generation and quality control, are clearly articulated, making the methodology easy to follow.

**Novelty and Soundness:**

* The paper addresses a pressing need in the research community for scalable, executable repository-level datasets, and the proposed RepoST method shows strong potential for continued scalability to meet future research demands.
* The dataset construction and quality control pipeline are meticulously designed, resulting in a high-quality dataset with rigorous filtering and debugging processes.
* Comprehensive experiments, including RFT analysis, human performance benchmarking, and docstring analysis, enhance the robustness and soundness of the proposed framework.

**Reasons To Reject:**

**Performance:**

* While the inclusion of RepoST-Train leads to some improvements, the overall gains remain relatively modest. The limited improvement on HumanEval is understandable given its algorithmic focus, but the marginal gains on RepoST-Eval raise questions about the dataset’s efficacy in training repo-level models. A more granular error analysis could provide valuable insights into the underlying challenges.

**Repo-Level Context:**

* The current implementation primarily focuses on local dependencies for context generation. However, generating code for repo-level functions may benefit from broader contextual information, such as similar or related functions across other files in the repository. Discussing how RepoST handles such contextual retrieval could further strengthen the framework.

**Cost Considerations:**

* The paper does not provide sufficient information regarding the cost of constructing RepoST, including API calls and manual verification efforts. A rough cost breakdown or discussion of resource allocation would be useful to assess the practical feasibility and scalability of the proposed pipeline.

---

> ### Author Response · Authors · 2025-06-03
>
> Thanks for your feedback and suggestions!
>
> &nbsp;
> ### Q1: Performance & Error analysis on Repo-level Code Generation
> We sampled 10 failure examples from RepoEval and manually checked the causes of error:
> * Instruction not clear: 2/10 (1/10 does not have docstrings)
> * package version mismatch (e.g., transformers): 1/10
> * problem with boundary cases: 4/10 (3/10 are not specified clearly)
> * correct functionality but wrong implementation: 1/10
> * wrong functionality: 1/10
> * format error (e.g., does not generate a function): 1/10
>
> Among the 10 failure cases, 6/10 are actually due to unclear instructions or unclear package versions, only 4/10 are actually incorrect. In other words, these 6/10 examples cannot even be solved by human. This is a known issue with the RepoEval dataset, where the intention of the functions cannot be fully entailed from the context.
>
> Thus, performance is calculated over the total set of examples, including many that are inherently unsolvable or ambiguous. We believe adjusting for this by considering only clearly solvable cases would highlight a more substantial gain from our method.
>
> We observe similar issues with our RepoST-Eval datasets and present a potential solution: using model-generated docstrings to improve clarity (see Sec. 5 for details).
>
> &nbsp;
> ### Q2: Repo-level context
> It is totally possible to use other types of context without even modifying the RepoST-Eval / RepoST-train datasets. Note that we provide the .zip file containing all the repos, we only need to mask the target function, retrieve relevant context for each function, and perform retrieval-augmented code generation. We listed it as a promising future work in the “ Conclusion and Future Works” section.
>
> &nbsp;
> ### Q3: Cost
> The cost of the RepoST-Eval dataset is 0.14 per example with GPT-4o. Note that the cost can be further reduced with cheaper and stronger models such as o3-mini.

---

> > ### Comment · Reviewer_dCkp · 2025-06-07
> >
> > Thank you for the clarification and the additional results. I’ll stick with my current score for now.

---

### Official Review · Reviewer_GgBH · 2025-05-13

**Rating:** 6
**Confidence:** 4
**Ethics Flag:** 1

**Summary:**

This paper presents REPOST, a scalable framework for constructing repository-level coding environments using sandbox-based testing to generate execution feedback for code generation models. Unlike traditional integration testing, REPOST isolates functions and their local dependencies, simplifying environment setup and greatly improving scalability.

**Questions To Authors:**

1. It would strengthen the paper by clearly differentiating your method's novelty from closely related recent works, especially R2E and SWE-Gym, emphasizing distinct conceptual or practical improvements.

2. How robust is the sandboxing method when dealing with complex external APIs or highly interactive repository structures? Could you provide more detailed analyses or examples of failure cases?

3. I appreciate the contribution of this work and took time to examine details of the dataset. Line 115 mentions that "by isolating the function and its local dependencies, we can execute the function by only installing the necessary packages." However, the dataset seems to lack a critical field specifying the exact "necessary packages" (and their versions) needed to run the sandbox (eval_script). Considering the problem of third-party API evolution, preserving this requirement information could be crucial for future reproducibility and usability.

**Reasons To Accept:**

1. Proposes a sandbox-based framework that effectively addresses the scalability challenges of constructing executable environments, improving over prior integration testing approaches.
2. Fully automates environment setup, significantly reducing manual effort and enhancing practical scalability.
3. Introduces thorough quality control measures, including iterative debugging and test coverage improvements, to ensure the dataset’s reliability and quality.

**Reasons To Reject:**

1. Assumes relatively simple local dependencies and API mocking; the impact of this assumption in complex real-world settings is under-explored.
2. The human study is based on a small sample size (20 examples for quality checks, 27 examples for solvability), limiting the strength of conclusions about dataset quality and difficulty.

---

> ### Author Response · Authors · 2025-06-03
>
> Thanks for your thoughtful feedback. We appreciate the time you took to review our paper in detail. Below, we address each of your concerns:
>
> &nbsp;
> ### Q1: Sandboxing on complex cases and more analysis of API mocking
> We agree that the complexity of generated examples is inherently limited by the capabilities of current models. This limitation applies to all automatic environment construction methods, including R2E.
>
> Furthermore, RepoST is still able to sandbox complex functions with complex dependencies. For instance, our most complex evaluation script is 1143 lines long and aggregates context from 8 different files.
>
> To better balance simple and complex functions during training, a possible direction is to generate a large pool of examples and apply downsampling. We leave a comprehensive analysis of this strategy to future work.
>
> &nbsp;
> ### Q2: The scale of human study is too small
> We acknowledge that the human study is limited in size due to budget constraints. Despite the small sample, the results were consistent and supported by strong inter-rater agreement. Specifically, Cohen's Kappa was 0.9179 for functionality checks and 0.7750 for test checks, indicating high reliability.
>
> &nbsp;
> ### Q3: Novelty compared to R2E & SWE-Gym
> The major difference between our method and SWE-Gym is that we present a fully automatic framework, while SWE-Gym requires huge human effort to set up the coding environments.
>
> Compared to R2E, which is also automatic, the major novelty is the design of “sandbox testing”. As introduced in the introduction section, R2E leverages integration testing, which requires setting up the repository environment. Setting up the repository environment is a difficult task, and suffers from low success rate when automated by an LLM. In comparison, we present sandbox testing that tests the target function in a separate script. With sandbox testing, the LLM only needs to install the external packages for the target function and its necessary dependencies (which is much fewer than the entire repository), and can also debug the evaluation script without breaking the original repository.
>
> With an automated framework with sandbox testing, our method achieves significantly better scalability than both SWE-Gym and R2E. Please refer to Table 1 for comparative statistics.
>
> &nbsp;
> ### Q3: Analysis of failure cases of API mocking
> We evaluated the success rate of functions with and without external APIs on RepoST-Eval:
> * With external APIs: 34/234 (14.5%)
> * Without external APIs: 262/809 (32.4%)
> The results are expected because in the former case, the model will need to construct the mock API classes from scratch.
>
> We manually checked the examples involving API mocking and discovered three common failure types: (1) the model does not create a mock class, or directly call the original external API in the evaluation script, (2) the mock class has compilation/runtime errors, and (3) the mock class has a different interface as the real external one, so the functionality of the sandboxed function has altered.
>
> It's worth noting that existing approaches like R2E avoid this problem entirely by filtering out such examples at the first step. To our knowledge, RepoST is the first method that actively explores evaluation script construction involving API mocking.
>
> &nbsp;
> ### Q4: the dataset seems to lack a critical field specifying the exact "necessary packages" (and their versions)
> We appreciate your effort in carefully reviewing the datasets. While we do not explicitly list required packages per example, all examples in our dataset can be executed within a common Python environment. We have released Docker images for both RepoST-train and RepoST-eval. Users can reproduce our results by pulling these images.
>
> The exact package versions can be checked by (1) building a docker container based on our image, and (2) inside the container, run `pip freeze > requirements.txt` to check the python and package environments.
>
> We believe this setup provides a practical and reproducible way to manage dependency information.

---

> > ### Comment · Reviewer_GgBH · 2025-06-10
> >
> > Thanks for providing the rebuttal. I have decided to retain my original assessment after reading it since my assessment is already positive.

---

### Official Review · Reviewer_qkFa · 2025-05-14

**Rating:** 7
**Confidence:** 4
**Ethics Flag:** 1

**Summary:**

REPOST addresses the challenge of repository-level code generation by automatically constructing executable coding environments using sandbox testing. Instead of attempting full integration testing (running original repository test suites, which requires building the entire repo environment), RepoST isolates each target function along with its dependencies into a self-contained script for unit testing. This sandboxing approach dramatically simplifies environment setup by avoiding complex external dependencies and install issues. Using this method, the authors curate RepoST-Train, a training set of 7,415 functions from 824 GitHub repositories, each paired with an auto-generated test suite. They also release RepoST-Eval (296 functions from 99 repos) for benchmarking models on realistic, multi-file coding tasks. Training a code language model on RepoST-Train with execution feedback yields performance gains (+5.5% Pass@1 on HumanEval, +3.5% on a prior repo-level eval). The paper evaluates 12 code generation models on RepoST-Eval where GPT-4 achieves only ~39.5%, indicating the difficulty of these tasks and room for improvement. Overall, RepoST’s novel sandbox testing paradigm enables scalable creation of repository-level code generation benchmarks.

**Questions To Authors:**

1. How well would the REPOST sandbox strategy generalize to tasks requiring modifications across multiple functions or files (beyond a single target function)? In other words, can this approach be extended to handle a multi-function feature implementation or bug fix, or is it inherently limited to one function at a time?

2. Did you observe any cases where the LLM-generated tests were incomplete or too lenient, allowing incorrect implementations to pass? More broadly, what’s the confidence that the generated tests fully capture the specification of the target function, and have you considered techniques (or metrics) to measure and improve the specification completeness of these tests?

**Reasons To Accept:**

1. The paper addresses a critical gap by moving to realistic repository-level code generation tasks, which is highly relevant for advancing code AI. The new REPOST-Eval benchmark is challenging (GPT-4 Pass@1 ~ 39%) and will drive research on more capable coding models.

2. It proposes a novel solution (sandbox testing) to automate environment construction. This approach is innovative compared to prior methods (r2e) and enables an order-of-magnitude more tasks (7.4k vs a few hundred) to be collected. The idea of isolating functions with mocks is a clever way to obtain execution feedback at scale.

3. The pipeline is well-designed with call graph dependency extraction, LLM-generated tests, and rigorous quality control (including LLM and human checks) to ensure each example is correct. High test coverage (~ 98%) indicates the generated tests are thorough. The paper provides strong evidence that training on execution feedback data improves models  (+5.5% Pass@1 on HumanEval) after fine-tuning with REPOST-Train.

4. Overall the paper is clearly written and structured, with detailed examples and an open-source release of code/data. The process is well-documented (prompts in appendix, etc.); the contribution is easy to understand.

**Reasons To Reject:**

1. RepoST focuses only on functions that can be tested in isolation. This excludes more complex “integration” tasks where multiple components must work together. Thus, the benchmark may not cover certain real-world scenarios (e.g. adding a feature touching multiple files). The claim of handling repository-level generation is somewhat restricted and limits the impact or generality of the approach. Also it looks like it is focused on python which makes the impact further limited

2. The pipeline heavily depends on GPT-4 for creating the dataset (merging code, writing mocks/tests). The reliance on closed-source AI for data generation might complicate replication.

3. Quality. The dataset’s automatic nature might still hide some incorrect assumptions or trivialized tasks. For example, some tests might not fully specify the problem (e.g., only one possible solution fits the tests but others are logically valid). Such issues could limit the dataset’s effectiveness or the correctness of evaluation. The model then is trained to reproduce possibly flawed logic.

4. Limited eval: it is well known that humaneval is saturated and contaminated with the latest model. The authors must present results on other eval datasets like livecodebench and bigcodebench.

---

> ### Author Response · Authors · 2025-06-03
>
> Thank you for your thoughtful and constructive feedback! We would like to address your concerns as follows:
>
> &nbsp;
> ### Q1: Generalize RepoST to tasks involving multiple components
> This is a good question. We did list it as one of the future directions in the “Conclusion and Future Works” section.
>
> In future work, we can generalize RepoST to multi-component feature implementation as follows:
> (1) In inference, we mask all components directly relevant to a specific feature. Then we apply models to generate code based on the partially masked repository context.
> (2) When generating the evaluation script, since we already aggregate all dependent components to the evaluation script, no changes are required in the sandboxing stage. In the post-checking stage, we need to check whether the behavior of all the components to generate remains the same. Also, in the test generation step, we need to prompt the LLM to generate tests for the entire feature, not just a single function. Similarly to our current design, the GT modules must pass all the generated test cases.
> (3) In evaluation, we copy all the generated components to the evaluation script, including the previously masked modules and all standalone functions and classes.
>
> Similarly, we can also generalize RepoST to issue-solving tasks as follows:
> (1) We extract all modified modules for fixing the bug (e.g., by extracting from an issue-solving PR)
> (2) Then we sandbox and generate tests for each module separately
> (3) In evaluation, we check whether the models’ generated patch passes the tests for all the modules
>
> The above extensions show that RepoST can be adapted beyond single-function generation tasks.
>
> &nbsp;
> ### Q2: Reliance on closed-source AI
> While we used GPT-4 to construct our initial dataset, RepoST is model-agnostic and can easily incorporate open-source LLMs (e.g., `deepseek-ai/DeepSeek-V3-0324`). Our recent experiments on SWE-Bench show that deepseek-v3 is even stronger than GPT-4o on issue solving, a specific repo-level coding task, making it a reasonable alternative model.
>
> &nbsp;
> ### Q3: The tests are not perfect, which may affect training data quality
> You’re absolutely right that LLM-generated tests may occasionally contain flaws. Our human study (Sec. 3.1) found that 1 out of 10 tests labeled "valid" by GPT-4o was actually incorrect.
>
> However, we want to highlight that even with imperfect tests, RFT training still improves model performance. For instance, even lenient tests can ensure that the target output function can compile, run, and produce outputs without errors. In other words, the model will learn to produce code with no runtime/compilation errors from those training examples.
>
> &nbsp;
> ### Q4: Cases where the LLM-generated tests were incomplete or too lenient and how to improve it
> As noted, in our human study (Sec. 3.1), 1 out of 10 tests labeled "valid" by GPT-4o was actually incorrect. In the only failure case, the goal of the target function is to edit some class attributes. Our failure case involves a test function that doesn’t reset the class attributes before calling the model-implemented function. Despite this, this test still filters out generations with compilation/runtime errors.
>
> To improve test quality, one way is to finetune an evaluator model to judge test correctness. Specifically, the training data can be collected using SWT-Bench’s method: (1) collect issue-solving PRs, (2) use an LLM to rollout candidate tests, and (3) label the generated test as correct/incorrect by whether the modified function in the PR can pass the test but its original implementation cannot.
>
> &nbsp;
> ### Q5: HumanEval is contaminated
> We agree that HumanEval is saturated and potentially contaminated in modern LLM training. However, we would like to emphasize that RepoEval is our primary benchmark. It is a repo-level dataset specifically tailored to the repo-to-function generation task. To our knowledge, RepoEval is not as saturated as HumanEval, and better reflects the complexities of real-world development.

---

### Decision · Program_Chairs · 2025-07-08

**Decision:**

Accept

**Comment:**

The paper presents RepoST, a novel framework for repository-level code generation that focuses on sandbox testing instead of traditional integration testing approaches. This work addresses the important challenge of scalable execution environment construction for code generation models by isolating target functions and their dependencies into self-contained scripts. The contribution is significant and timely given the growing interest in repository-level code generation capabilities both from evaluation as well as training with execution feedback perspective. The authors have developed both a training dataset (RepoST-Train with 7,415 functions) and an evaluation benchmark (RepoST-Eval with 296 functions), showing that training with execution feedback from their system yields meaningful performance improvements across benchmarks. While the approach has limitations in handling complex multi-component tasks, the paper is clearly written, technically sound, and provides a valuable contribution to the field that enables more realistic evaluation and training on repository-level code tasks. While authors don't explore it in the paper, I believe the proposed sandbox testing env can be used for reinforcement learning with verified reward, a key direction for community.

Pros
- Addresses a critical gap in code generation by focusing on realistic repository-level code tasks beyond simple function-level generation
Introduces an innovative sandboxing approach that overcomes scalability issues of previous integration testing methods
- Creates substantially larger datasets (7,415 functions) compared to prior work through automation
- Demonstrates concrete improvements when fine-tuning models with execution feedback (+5.5% on HumanEval, +3.5% on RepoEval)
- Well-designed pipeline with thorough quality control measures and high test coverage (~98%)
- Open-source release of code and data enhances reproducibility and community impact

Cons
- Limited to functions that can be tested in isolation, excluding more complex real world scenarios.
- Relatively small human evaluation studies (20-27 examples) to validate dataset quality
- Currently focused primarily on Python, limiting language coverage